# Personalized antibiograms for machine learning driven antibiotic selection

Conor K. Corbin [1✉], Lillian Sung[2], Arhana Chattopadhyay[1], Morteza Noshad[1], Amy Chang [3], Stanley Deresinksi[3], Michael Baiocchi[1] & Jonathan H. Chen [1]

## Abstract

**Background** The Centers for Disease Control and Prevention identify antibiotic prescribing stewardship as the most important action to combat increasing antibiotic resistance. Clinicians balance broad empiric antibiotic coverage vs. precision coverage targeting only the most likely pathogens. We investigate the utility of machine learning-based clinical decision support for antibiotic prescribing stewardship.

**Methods** In this retrospective multi-site study, we developed machine learning models that predict antibiotic susceptibility patterns (personalized antibiograms) using electronic health record data of 8342 infections from Stanford emergency departments and 15,806 uncomplicated urinary tract infections from Massachusetts General Hospital and Brigham & Women's Hospital in Boston. We assessed the trade-off between broad-spectrum and precise antibiotic prescribing using linear programming.

**Results** We find in Stanford data that personalized antibiograms reallocate clinician antibiotic selections with a coverage rate (fraction of infections covered by treatment) of 85.9%; similar to clinician performance (84.3% $p = 0.11$). In the Boston dataset, the personalized antibiograms coverage rate is 90.4%; a significant improvement over clinicians (88.1% $p < 0.0001$). Personalized antibiograms achieve similar coverage to the clinician benchmark with narrower antibiotics. With Stanford data, personalized antibiograms maintain clinician coverage rates while narrowing 69% of empiric vancomycin+piperacillin/tazobactam prescriptions to piperacillin/tazobactam. In the Boston dataset, personalized antibiograms maintain clinician coverage rates while narrowing 48% of ciprofloxacin to trimethoprim/ sulfamethoxazole.

**Conclusions** Precision empiric antibiotic prescribing with personalized antibiograms could improve patient safety and antibiotic stewardship by reducing unnecessary use of broad-spectrum antibiotics that breed a growing tide of resistant organisms.

## Plain language summary

Antibiotic resistance is an increasing threat to public health. The World Health Organization estimates that 700,000 people die annually due to antibiotic resistant infection. By 2050 the annual death toll is expected to reach 10 million. The Centers for Disease Control and Prevention list the importance of appropriate prescribing of antibiotics as the number one action advised to reduce the spread of resistant bacteria. When selecting appropriate antibiotics, clinicians aim to maximize the likelihood that individual patients will respond whilst limiting the use of options that have action against a large number of different bacteria. Overuse of valuable wide acting antibiotics can increase the rate at which bacteria develop resistance to them. Here we show that machine learning models that predict antibiotic susceptibility have the potential to guide clinicians when choosing antibiotics in a way that maintains or improves patient safety while reducing the overall use of wide acting antibiotics.

[1] Center of Biomedical Informatics Research, Stanford University, Stanford, CA, USA. [2] Division of Haematology/Oncology, The Hospital for Sick Children, Toronto, ON M5G1X8, Canada. [3] Medicine and Infectious Diseases, Stanford Medicine, Stanford, CA, USA. ✉email: ccorbin@stanford.edu

The World Health Organization (WHO) estimates that 700,000 people already die annually due to antibiotic resistant infections, and expects this number to exceed 10 million per year by 2050[1]. Increasing antibiotic resistance is a natural and inevitable consequence of regular antibiotic use, raising the looming threat of a post-antibiotic era that could cripple routine medical care with higher infection-related mortality and costs of care[2–5].

The Centers for Disease Control and Prevention (CDC) identify improving antibiotic prescribing through antibiotic stewardship as the most important action to combat the spread of antibiotic resistant bacteria[6]. For example, sixty percent of hospitalized patients receive antibiotics despite the fact that half of antibiotic treatments are inappropriate—meaning antibiotic use was unwarranted, the wrong antibiotic was given, or the antibiotic was delivered with wrong dose or duration[7]. A key challenge is that antibiotics must often be prescribed empirically, before the identity of the infecting organism and antibiotic susceptibilities are known. Microbial cultures are the definitive diagnostic tests for this information, but may take days to confirm final results, far too long to delay initial therapy[8].

Broad-spectrum antibiotics help ensure coverage of a range of organisms that would lead to rapid clinical deterioration if left untreated[9,10]. Yet, it is precisely the excessive use of antibiotics that increases drug resistant organisms[11]. Overuse of broad-spectrum antibiotics can thus have severe immediate and indirect consequences ranging from increasing antibiotic resistance to drug-specific toxicities and secondary infections such as *Clostridioides difficile colitis*[12–14].

Existing standards of care for selecting empiric antibiotics involve referring to clinical practice guidelines combined with knowledge of institution-specific antibiograms—an annual report from an institution's microbiology lab that tracks the most common organisms isolated by microbial cultures and the percentages that were found susceptible to different antibiotics[15–17]. An institution's antibiogram might report for example that 1000 *Escherichia Coli* were isolated in the prior year, and that 98% were susceptible to meropenem, while only 89% were susceptible to ceftriaxone. These approaches may not consider many or any patient-specific features. Microbial culture results found within the electronic health record can be used to objectively measure not only whether chosen antibiotics were appropriate, but if alternatives would have sufficed. Here we hypothesize that the standard of care may benefit from machine learning-based clinical decision support for personalized treatment recommendations.

The development of computerized clinical decision support for antibiotic prescribing stems back decades to the likes of MYCIN and Evans et al.—rule-based systems that guide clinicians through empiric antibiotic selection[18,19]. Though promising, neither system was widely adopted by clinicians as they were not easily integrated into their medical workflow or adaptable to constantly evolving local antibiotic resistance patterns[20]. With modern day hospital IT and electronic medical record software it is now possible to integrate clinical decision support into medical workflows and dynamically train models with real world clinical data streams[21].

Literature concerning modern day data-driven approaches to antibiotic decision support fall into two distinct categories. One category of studies predict infection status at the time microbial cultures were ordered, offering promising consideration for when antibiotics are needed at all[22–24]. Limitations in most of these prior studies is that positive microbial culture results were used as a proxy for the outcome of infection, despite their being both false positive and false negative microbial cultures with respect to an actual clinical infection. Moreover, these studies do not address the question of which antibiotics should have been administered.

The second category of studies predict antibiotic susceptibility results for positive microbial cultures[25–28]. These studies address the challenge of selecting the right antibiotic. Antibiotic prescribing policies that leverage machine learning predictions were simulated and benchmarked against retrospective clinician prescribing and suggested improved performance. Optimizing patient coverage rates, however, is only one important objective that could be naively addressed by prescribing maximally broad antibiotics to all patients without consideration for adverse effects on the individual or population. Further critical research needs to systematically evaluate the trade-off between maximizing antibiotic coverage across a population of patients and minimizing broad-spectrum antibiotic use.

In a previous work we demonstrated that machine learning models could predict antibiotic susceptibility results when conditioned on microbial species[29]. We examined precision-recall curves of these models and highlighted thresholds that separated subgroups of patients with probability of coverage with narrower-spectrum antibiotics equal to antibiogram values of broader-spectrum antibiotics. Here we substantially extend our work on personalized antibiograms to generalize beyond species identity, introduce a linear programming optimization framework to simulate optimal antibiotic allocations across a set of patients, conduct a sensitivity analysis to estimate model performance on patients with negative microbial cultures, and assess the generalizability of our findings with data from an external site. Specifically, our objective in this study is the following.

We (1) train and evaluate personalized antibiograms—machine learning models that use electronic health record data to predict antibiotic susceptibility results; (2) evaluate the performance of antibiotic selections informed by personalized antibiograms relative to selections made by clinicians; and (3) systematically evaluate the trade-off in performance when fewer broad-spectrum antibiotics are selected across a population of patients. We complete this objective using a cohort of patients who presented to Stanford emergency departments between 2009 and 2019 and then replicate our process on an external cohort of patients who presented to the Massachusetts General Hospital and Brigham & Women's Hospital in Boston between 2007 and 2016.

In our Stanford cohort we find that personalized antibiograms are able to reallocate antibiotic selections made by clinicians with a coverage rate (defined as the fraction of infections covered by the antibiotic selection) of 85.9%, similar to the clinician coverage rate (84.3%, $p = 0.11$). We find in the Boston data that personalized antibiograms reallocate antibiotic selections with coverage rate of of 90.4%—significantly higher than the coverage rate clinicians achieve (88.1% $p < 0.0001$). In the Stanford data we find that antibiotic selections guided by personalized antibiograms achieve a coverage rate as good as the real world clinician prescribing rates while narrowing 69% of the vancomycin + piperacillin/tazobactam selections to piperacillin/tazobactam, 40% of piperacillin/tazobactam prescriptions to cefazolin, and 21% of ceftriaxone prescriptions to ampicillin. In the Boston data we find that personalized antibiograms can replace 93% of the total ciprofloxacin prescriptions with nitrofurantoin without falling below the real world coverage rate. Similarly 48% of the total ciprofloxacin and 62% of nitrofurantoin prescriptions can be exchanged with trimethoprim/sulfamethoxazole.

## Methods

**Data sources**. We used the STAnford Research Repository (STARR) clinical data warehouse to extract de-identified patient medical records[30]. STARR contains electronic health record data collected from over 2.4 million unique patients spanning 2009–2021 who have visited Stanford Hospital (academic medical

center in Palo Alto, California), ValleyCare hospital (community hospital in Pleasanton, California) and Stanford University Healthcare Alliance affiliated ambulatory clinics. We included patient encounters from both the Stanford and ValleyCare emergency departments. Structured electronic health record data include patient demographics, comorbidities, procedures, medications, labs, vital signs, and microbiology data.

STARR microbiology data contain information about microbial cultures ordered within the hospital, emergency departments, and outpatient clinics. Our microbiology data included source of culture, order timestamp, and result timestamp. Microbial culture data also included resulting organism name and antibiotic susceptibility pattern, which indicated whether each organism was either susceptible, intermediate, or resistant to a set of tested antibiotics. Microbiology data collected from ValleyCare and Stanford emergency departments were analyzed at separate microbiology labs. Both follow standardized national procedures to measure antibiotic susceptibility as defined by the Clinical & Laboratory Standards Institute (CLSI)[31]. Our study was approved by the institutional review board of the Stanford University School of Medicine. Project-specific informed consent was not required because the study was restricted to secondary analysis of existing clinical data.

We replicated the analysis on electronic medical record data of patients from Massachusetts General Hospital and the Brigham and Women's Hospital in Boston, MA—a dataset made available through Physionet[28,32].

**Cohort definitions**. The unit of observation in this analysis was a patient-infection. In the Stanford data, analysis was restricted to patients who presented to the emergency department with infection between January 2009 and December 2019. We included patients 18 years or older and patients who required hospital admission. We further restricted the cohort to patients where an order for at least one positive blood, urine, cerebral spinal fluid or fluid microbial culture; and, at least one order for intravenous or intramuscular antibiotics were placed in the first 24 h after presentation to the emergency department. We excluded observations where antibiotics or microbial cultures had been ordered within the 2 weeks prior to the presentation to the emergency department. We incorporated admissions with negative cultures in a sensitivity analysis. Figure 1 illustrates the flow diagram of patient evaluation, reasons for exclusions and number included in our study tabulating both the number infections and number of unique patients. In the Boston dataset the unit of observation was similarly a patient-infection. Analysis was restricted to uncomplicated urinary tract infections, as described in Kanjilal et al.[28].

Observations between the years 2007 and 2016 were included in the study.

**Labelling infections for personalized antibiogram models**. Using Stanford data we trained 12 binary machine learning models to estimate the probability that common antibiotic selections would provide activity against infections at the point in time empiric antibiotics were chosen. An antibiotic selection was said to provide activity against a patient's infection if all microbial organisms that grew in the patient's microbial cultures were listed as susceptible to at least one of the antibiotics in the selection. While microbial cultures growing *Coagulase-Negative Staphylococci* sometimes represent true infections warranting antibiotic treatment, we excluded *Coagulase-Negative Staphylococci* cases as they frequently represent non-infectious contaminants. We trained models for eight commonly administered single antibiotic choices (vancomycin, piperacillin/tazobactam, cefepime, ceftriaxone, cefazolin, ciprofloxacin, ampicillin and meropenem) and four combination therapies (vancomycin + piperacillin/tazobactam, vancomycin + cefepime, vancomycin + ceftriaxone, and vancomycin + meropenem). We defined our prediction time to be the time at which the first intravenous or intramuscular antibiotic was ordered for the patient following admission to the emergency department. Using Boston data, we similarly trained personalized antibiogram models that predicted whether four antibiotics commonly administered for urinary tract infection (trimethoprim/sulfamethoxazole, nitrofurantoin, ciprofloxacin, and levofloxacin) would provide activity against the target infection.

Not all antibiotics were tested against all organisms in the microbiology lab which resulted in missing labels for some of our observations. Antibiotics are only tested if they could plausibly be active against a specific organism. We received consultation from the Stanford microbiology lab to generate a set of rules to impute missing labels. For example, our imputation rule assumed that *Pseudomonas aeruginosa* would be resistant to ceftriaxone and cefazolin; Gram negative rods would be resistant to vancomycin; and *Streptococcus agalactiae* is susceptible to cephalosporins[33–35]. Antibiotic susceptibility was also inferred from observed results of related antibiotics. For example, if an organism was susceptible to a first-generation cephalosporin, it was assumed that it would also be susceptible to a second, third or fourth-generation cephalosporin[36].

**Feature engineering**. Using Stanford data, a feature matrix was constructed with data in the EHR with timestamps up until the prediction time. Though observations were restricted to infections that required hospital admission, features were constructed based

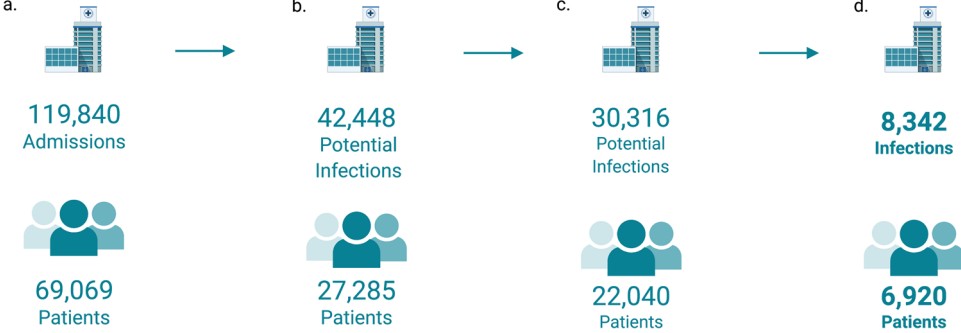

**Fig. 1 Study cohort selection. a** 119,840 hospital admissions corresponding to 69,069 unique adult patients admitted from Stanford emergency rooms between 2009 and 2019 were initially examined for inclusion. **b** 42,448 admissions had a microbial culture and intravenous or intramuscular empiric antibiotic order placed within the first 24 h of the encounter. **c** Admissions were excluded if microbial cultures had been ordered in the 2 weeks leading up to the encounter. **d** Admissions resulting in negative microbial cultures were excluded in the primary analysis, leaving 8342 infections from 6920 unique patients.

on data from all forms of available medical encounters (including ex: prior primary care visits). We used a bag of words featurization technique similar to Rajkomar et al. to construct our feature matrix[37]. Categorical features included diagnosis codes (ICD9 and ICD10 codes), procedure orders, lab orders (including microbiology lab orders), medication orders, imaging orders and orders for respiratory care. For prior microbial culture results, categorical features were constructed based on the antibiotic susceptibility pattern of extracted isolates. Bag of words feature representations for each admission were generated such that the value in each column was the number of times that feature was present in the patient's record during a pre-set look back window. For diagnosis codes, the look back window was defined as the entire medical history. For other categorical features, the look back window was defined as the year leading up until prediction time. If a feature was not present for a patient (for example a diagnosis code was never assigned or a lab test was never ordered), the value in the corresponding column was zero. This allowed us to implicitly encode missing values into our representation without having to impute our data.

Numeric features included lab results and vital signs from flowsheet data. We binned the values of each unique numerical feature into categorical buckets based on the decile cutoffs of their distributions in the training set. We used the training set to identify thresholds for each decile in a feature's distribution and then applied these thresholds to patients in our validation and test set to prevent information leakage. To create the bag of words representation, we created columns in our feature matrix where the corresponding value represented the number of times the feature with a value in a particular decile was observed within a look back window. For lab results and vital signs, the look back window was 14 days prior to prediction time. Features were not standardized, and rather left as counts.

In addition to these categorical and numeric features, we included patient demographics (age in years, sex, race, and ethnicity), insurance information, and institution (Stanford or ValleyCare). Sex, race, ethnicity, insurance payer, and institution were one-hot encoded. In total, the sparse feature matrix contained 43,220 columns.

With Boston data, a feature matrix was generated as in Kanjilal et al.[28] Features included prior microbiology data, antibiotic exposures, comorbidities, procedures, lab results and patient demographics. The total number of features used in this portion of the analysis was 788.

**Training and model selection procedure**. With Stanford data we split the cohort by year into training (2009–2017), validation (2018), and test (2019) sets to mimic distributional shifts that occur with deployment[38]. This is particularly important so that we can take into consideration changes in the data generating process (resistance patterns and medical practice) that occur over time when estimating model performance. We did not re-weigh or re-sample our training data according to class balance in an attempt to preserve model calibration on our test set[39].

We selected from four model classes: L1 (lasso) and L2 (ridge) logistic regressions, random forests, and gradient boosted trees. These were specifically chosen so that we could search over model classes with different biases and variances. The L1 and L2 penalized logistic regressions assume the outcome is a linear function of the features, are less flexible, but also less prone to overfitting the data. The random forests and gradient boosted trees can model nonlinear interactions between features and outcomes, are more flexible, but more prone to overfitting. Random forests perform inference by averaging predictions from a collection of trees, and gradient boosted trees perform inference

by summing the predictions of a collection of trees that each fit the residuals at the prior boosting round[40].

The training and model selection procedure we used for the logistic regressions and random forest is as follows. First, hyperparameters for each model class were selected by performing a stratified $k = 5$ fold cross validation grid search over the training set. Hyperparameters that led to the highest mean area under the receiver operating characteristic curve (AUROC) were selected for each model class. We then fit the final model for each model class using the entire training set and evaluated each on the validation set. The best model class was chosen by selecting the model with the highest AUROC in the validation set. After choosing the best model class, hyperparameters were re-tuned on the combined training and validation set using a stratified $k = 5$ fold cross validation grid search.

The training procedure was altered for the gradient boosted tree models so that we could regularize with early stopping. The training procedure was as above except that for each model fit, 5% of each training fold was held out and used as an additional validation set for early stopping. We set the maximum number of boosting iterations to 1000 and a tolerance of ten boosting rounds for the early stopping criteria.

The final model was then trained using the combined training and validation set and final performance was evaluated on the test set. The logistic regressions and random forest models were fit using the sci-kit learn python package[41]. The gradient boosted tree models were fit using the lightgbm python package[42]. We computed the area under the receiver operating characteristics curve (AUROC) and average precision, with 95% confidence intervals estimated by bootstrapping the test set 1000 times[43]. We list all tested hyperparameter configurations in Supplementary Note 1.

The Boston data was split into training (2007–2013) and test (2014–2016) sets by time (as in Kanjilal et al.). The optimal model class and hyperparameter setting for each of the four binary models was chosen with a $k = 5$ fold cross validation grid search over the training set[28].

**Optimizing antibiotic selection with personalized antibiograms**. We used the out of sample predicted probabilities from each of our binary classifiers to optimize antibiotic selections across patients in the test set and benchmarked against: (1) random antibiotic selections and (2) the observed clinician selections. Clinician performance was measured by extracting which antibiotics were administered to patients using information stored in the medication administration records. In the Stanford data, we restricted this analysis to admissions in the test set where one of the twelve antibiotic selections we trained models for were administered. In the Boston data, analysis was similarly restricted to patients who were prescribed one of the four antibiotics we trained personalized antibiogram models for.

The optimized antibiotic selections were generated by solving a constrained optimization formulation using linear programming[44]. For each admission we selected an antibiotic option that maximized the predicted probability of choosing an antibiotic listed as susceptible subject to the constraints that (1) only one antibiotic option could be selected for a given patient infection and (2) the total number of times certain antibiotic options were selected across patients matched a fixed budget. In the initial simulation, this budget was defined to be the number of times particular antibiotic choices were actually administered by clinicians in the real-world data. Thus, if ceftriaxone was allocated 100 times in the data, the optimizer was similarly forced to allocate ceftriaxone 100 times. In further simulations, these budget constraints were perturbed to empirically estimate the trade-off between maximally selecting

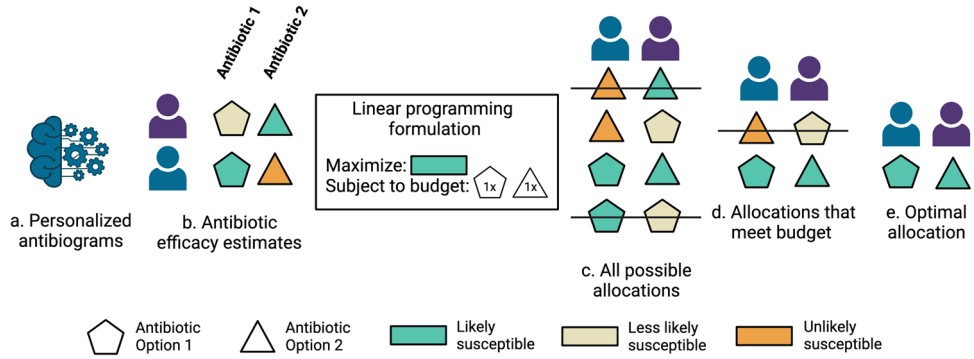

**Fig. 2 Optimizing antibiotic selections with linear programming.** Patient feature vectors are ingested by personalized antibiogram models (**a**) to produce antibiotic efficacy estimates (**b**). Each patient in the test set receives a predicted probability of efficacy for each antibiotic. In this illustration, pentagons refer to one antibiotic option and triangles refer to another. Green indicates the antibiotic option is likely to cover the patient, orange indicates the antibiotic is unlikely to cover the patient. A linear programming objective function is specified with a set of constraints that limit how frequently certain antibiotics can be used. Here the objective function specifies to maximize the total predicted antibiotic efficacy (green) across the two patients subject to the constraint that each antibiotic option is only used once. **c** Depicts all possible antibiotic allocations color coded by patient specific antibiotic efficacy estimates produced by personalized antibiograms. Antibiotics allocations are only considered (**d**) if they meet the constraints of the linear programming formulation. The antibiotic allocation that maximizes the total predicted efficacy across the set of patients (**e**) is chosen.

antibiotics with activity against a patient's infection and reducing the use of broad-spectrum antibiotics. An illustrative example of how patient feature vectors are converted into antibiotic assignments is shown in Fig. 2.

In technical detail, let $N$ be the number of admissions in our held out test set and $M$ be the total number of antibiotic selections. Let $S \in R^{N \times M}$ be a matrix of binary variables indicating antibiotic options, where $s_{ij} = 1$ represents that the $jth$ antibiotic option is allocated to the patient in the $ith$ admission. Let $\Phi \in R^{N \times M}$ be a matrix of out of sample probability estimates from our machine learning models, specifically $\phi_{ij}$ represents the predicted probability that the patient in the $ith$ admission would be covered by antibiotic option $j$. Finally, let $K_j$ be the total number of times the $jth$ antibiotic must be used over the set of our $N$ admissions (the budget parameters). For our initial simulation we let the budget parameters match the number of times clinicians actually allocated the $jth$ antibiotic selection over the set of $N$ admissions. Our problem formulation specified below was implemented using the PuLP python package and solved with the CBC solver[45] [Eq. 1].

$$\text{maximize } S \quad \sum_{i=1}^{N} \sum_{j=1}^{M} \phi_{ij} s_{ij}$$
$$\text{subject to} \quad \sum_{j=1}^{M} s_{ij} = 1 \quad i = 1, ..., N \tag{1}$$
$$\sum_{i=1}^{N} s_{ij} = K_j \quad j = 1, ..., M$$

**Sensitivity analysis.** We performed a sensitivity analysis to estimate model performance on the full deployment population, including patients with negative microbial culture results. This is a important because at prediction time, whether a microbial culture will return positive is unknown. Further, negative microbial cultures do not preclude infection at a site not tested. Some patients with negative microbial cultures will have a latent undetected infection with an antibiotic susceptibility profile (set of labels) that goes unobserved. This can skew model performance estimates if patients with censored labels have a covariate distribution different from those with observed labels. To address this, we (1) constructed an electronic phenotype to identify patients with negative microbial cultures that truly lacked infection, (2) re-trained a new set of personalized antibiogram prediction models that include patients flagged by the electronic phenotype and (3) used inverse probability weighted estimates of AUROC to evaluate performance on the deployment population,

the union of patient admissions with positive and negative microbial cultures.

*Electronic phenotype.* We created a rule based electronic phenotype that when applied to the set of patients in our cohort with negative microbial cultures attempted to extract instances where patients were truly uninfected. We created a strict phenotype, prioritizing positive predictive value over sensitivity. Patients were labelled as uninfected during the admission in question if the following was all true. None of the microbial cultures ordered within 2 weeks of the admission returned positive. As in the prior labelling scheme microbial cultures that grew only *Coagulase-negative Staphylococci* were considered negative. Antibiotics were either never administered, or they were stopped within 24 h of them starting. Antibiotics were not restarted for an additional 2 weeks if they were stopped. No ICD codes related to bacterial infection were associated with the hospital admission (see Supplementary Note 2). The patient did not die during the admission.

**Updated labelling schema.** Applying the above electronic phenotype to patients with negative microbial cultures resulted in a cohort of patient admissions broken down into three distinct buckets. Bucket 1 included patient microbial cultures that returned positive. Antibiotic susceptibility testing was performed and we observed their class label. This bucket is the set of patient infections included in our primary analysis. Bucket 2 included admissions whose microbial cultures returned negative and were flagged by our electronic phenotype indicating lack of infection. We observed these class labels, which we define as positive for each prediction task because lack of infection indicates the patient would have been covered by any antibiotic selection. Bucket 3 included admissions whose microbial cultures were negative and were not flagged by our electronic phenotype. Patients in this bucket may or may not have been infected. We did not observe their class labels. These three buckets are illustrated in Fig. 3.

We included patients in bucket 2 into our model training and evaluation procedure by adopting the following altered labelling schema. The labelling schema was as before except all patients in bucket 2 were assigned a positive label for every antibiotic. Specifically, for each of the twelve antibiotic options, a positive label was assigned if the admission resulted in a positive microbial culture (bucket 1) and the resulting organism(s) was susceptible

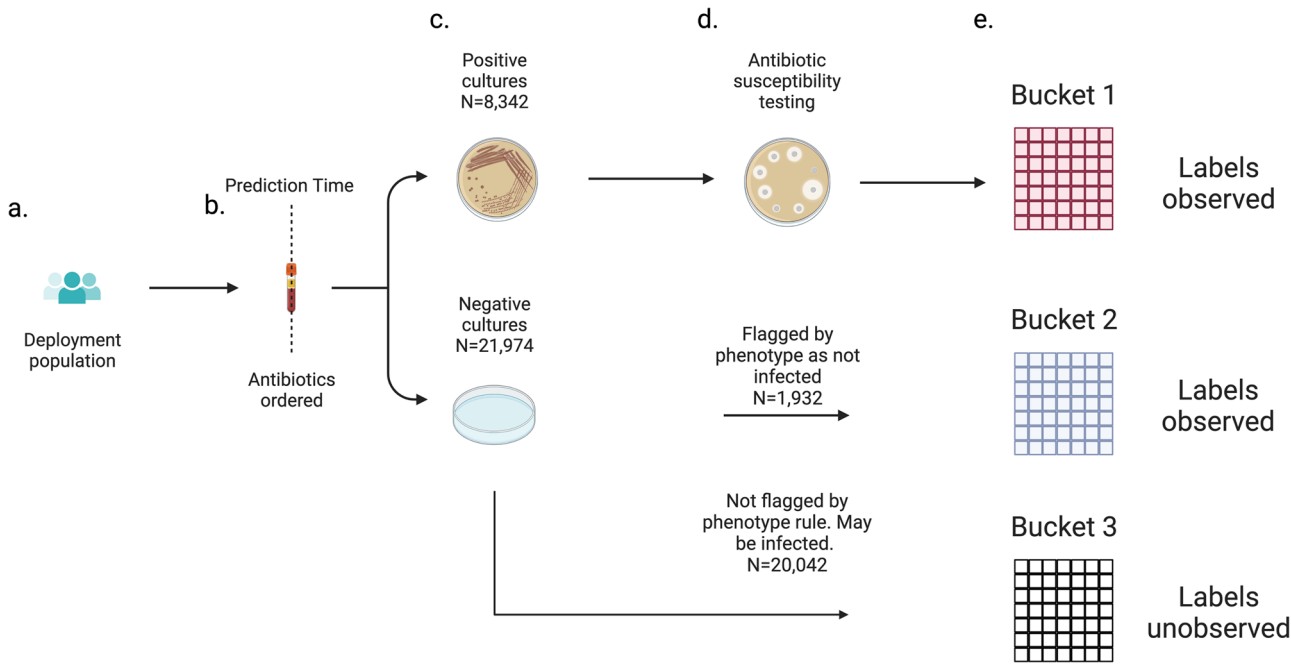

**Fig. 3 Three buckets of observations in the deployment population. a** The deployment population is the set of patients that would trigger personalized antibiogram model predictions in a deployment scenario. **b** Prediction time is defined as the time the empiric antibiotic order is placed. **c** After prediction time cultures can go on to have a positive or negative result. **d** If cultures are positive, antibiotic susceptibility testing is performed. If negative, our electronic phenotype flags patients who with high likelihood lacked a clinical infection that warranted antibiotics. **e** Three buckets of observations. Patients landing in Bucket 1 or 2 have observed labels in the labelling scheme defined in the sensitivity analysis. Patients landing in Bucket 3 have labels that go unobserved.

to the antibiotic, or the admission resulted in negative cultures and was flagged by the electronic phenotype (bucket 2). A negative label was assigned if the admission resulted in a positive microbial culture (bucket 1) and a resulting organism(s) was not susceptible to to the antibiotic.

Models were trained using patient admissions in buckets 1 and 2. The covariate distribution of patient admissions in bucket 3 was used to estimate model performance on the full deployment population (patients in buckets 1, 2, and 3) using inverse probability weighting.

**Inverse probability weighting to estimate model performance on the deployment population**. We used inverse probability weighting to account for patient admissions whose class labels we did not observe (bucket 3) in our estimates of model performance (AUROC). In a theoretical deployment scenario, we would deploy our model on a population of patient admissions that include the union of buckets 1, 2, and 3. If the covariate distribution of patients admissions in buckets 1 and 2 differs from the covariate distribution of patient admissions in bucket 3 and our models perform better or worse in regions of this covariate distribution that are more common for patient admissions in buckets 1 and 2, then we run the risk of over or underestimating how well our models would perform in deployment.

To estimate performance on a population of patients that includes a set of patients whose labels we do not observe, we weigh each patient admission whose label we do observe (bucket 1 and 2) by the inverse probability of observing it. We obtain this probability by fitting a binary random forest classifier (using the same feature matrix and index time as the personalized antibiogram models) to predict whether the patient admission would land in bucket 1 and 2, or bucket 3. The inverse probability weighted estimates of sensitivity and specificity for each of our 12 models are shown below. $\hat{f}(x^i)$ is the predicted probability from a

personalized antibiogram model for patient admission $i$, $\hat{P}(Obs^i = 1|X = x^i)$ is the predicted probability of observing patient admission $i$'s class label, and $t$ is the probability cut-off threshold to map predicted probability to a predicted class label. The inverse probability weighted ROC curve and area under it can be estimated by varying the probability cut-off thresholds of these estimators [Eqs. 2–3].

$$\widehat{Sensitivity}_{IPW} = \frac{1}{W_P} \sum_{i:Y^i=1} \frac{\mathbb{1}[\hat{f}(x^i) \ge t]}{\hat{P}(Obs^i = 1|X = x^i)},$$
$$W_P = \sum_{i:Y^i=1} \frac{1}{\hat{P}(Obs^i = 1|X = x^i)} \tag{2}$$

$$\widehat{Specificity}_{IPW} = \frac{1}{W_P} \sum_{i:Y^i=0} \frac{\mathbb{1}[\hat{f}(x^i) < t]}{\hat{P}(Obs^i = 1|X = x^i)},$$
$$W_P = \sum_{i:Y^i=0} \frac{1}{\hat{P}(Obs^i = 1|X = x^i)} \tag{3}$$

**Statistics and reproducibility**. Coverage rates of random antibiotic selection in the two cohorts were statistically compared to coverage rates achieved with personalized antibiograms with one-sided permutation tests[46]. Specifically, an empirical distribution of random coverage rates was created by repeatedly randomly re-assigning antibiotic selections to different patient-infections 10,000 times. A pvalue was calculated by taking the fraction of coverage rates in this empirical distribution that equaled or exceeded the coverage rate achieved with personalized anti-biograms. If no value in the empirical distribution equaled or exceeded the observed value, the pvalue was reported as $p < 0.0001$. The clinician coverage rate was compared to the personalized antibiogram coverage rate using a similar procedure, except that the empirical distribution of clinician coverage rates

**Table 1 Stanford cohort demographics grouped by train test split.**

| Description | Category | Dataset Test (2019) | Train + Validation (2009–2018) |
|---|---|---|---|
| n | Total | 1320 | 7022 |
| Emergency department, n (%) | Stanford ED | 855 (64.8) | 6669 (95.0) |
| | Valley Care ED | 465 (35.2) | 353 (5.0) |
| Age, mean (SD) | | 70.4 (17.2) | 67.5 (17.3) |
| Sex, n (%) | Female | 793 (60.1) | 4171 (59.4) |
| | Male | 527 (39.9) | 2851 (40.6) |
| Race, n (%) | White | 757 (57.3) | 3937 (56.1) |
| | Other | 251 (19.0) | 1411 (20.1) |
| | Asian | 201 (15.2) | 937 (13.3) |
| | Black | 69 (5.2) | 464 (6.6) |
| | Pacific Islander | 30 (2.3) | 206 (2.9) |
| | Unknown | 7 (0.5) | 40 (0.6) |
| | Native American | 5 (0.4) | 27 (0.4) |
| Ethnicity, n (%) | Non-Hispanic | 1117 (84.6) | 5823 (82.9) |
| | Hispanic/Latino | 195 (14.8) | 1169 (16.6) |
| | Unknown | 8 (0.6) | 30 (0.4) |
| Language, n (%) | English | 1112 (84.2) | 5743 (81.8) |
| | Non-English | 208 (15.8) | 1279 (18.2) |
| Insurance Payer, n (%) | Medicare | 651 (49.3) | 3805 (54.2) |
| | Other | 615 (46.6) | 2987 (42.5) |
| | Medi-Cal | 54 (4.1) | 230 (3.3) |

was generated by performing a stratified bootstrap (stratified by antibiotic selection) 10,000 times.

**Reporting summary**. Further information on research design is available in the Nature Research Reporting Summary linked to this article.

## Results

**Cohort description**. The Stanford cohort includes $N = 8342$ infections from 6920 adult patients who presented to the Stanford (academic medical center) emergency department or ValleyCare (affiliated community hospital) emergency department in Pleasanton, California between 2009 and 2019 and were subsequently admitted to the hospital for an infection. Some patients in our dataset have multiple instances of infection spanning their medical records. We include patients with multiple infections to mimic the deployment scenario where models would perform inference on patients with new infections who had prior infections seen during training. Demographic information is listed in Table 1, stratified by whether the infection is part of the training or test set. Table 2 demonstrates the most frequently isolated organisms stratified by anatomical source of the microbial culture.

**Personalized antibiograms**. We train binary machine learning models (personalized antibiograms) using structured electronic health record data to estimate the probability that infections would be susceptible to 12 common empiric antibiotics choices (four of which were combination therapies). Labels are derived from the antibiotic susceptibility reports of the microbial cultures. Observations are assigned a positive label when organisms isolated are deemed susceptible (not intermediate or resistant) to the respectively tested antibiotics based on microbiology lab standards.

Our dataset is split by time into training, validation and test sets containing $N_{train} = 5804$ patient-infections from 2009 to

**Table 2 Most frequently isolated species grouped by microbial culture type and emergency department.**

| Emergency department | Culture type | Organism | Infections |
|---|---|---|---|
| Stanford ED | Blood culture | Escherichia coli | 1031 |
| | | Staphylococcus aureus | 585 |
| | | Klebsiella pneumoniae | 318 |
| | | Enterococcus faecalis | 159 |
| | | Streptococcus agalactiae (group b) | 131 |
| | Urine culture | Escherichia coli | 2927 |
| | | Enterococcus species | 877 |
| | | Klebsiella pneumoniae | 653 |
| | | Proteus mirabilis | 299 |
| | | Pseudomonas aeruginosa | 268 |
| | Other fluid culture | Staphylococcus aureus | 127 |
| | | Escherichia coli | 83 |
| | | Streptococcus anginosus group | 56 |
| | | Klebsiella pneumoniae | 45 |
| | | Enterococcus faecium | 28 |
| Valley care ED | Blood culture | Escherichia coli | 98 |
| | | Staphylococcus aureus | 49 |
| | | Klebsiella pneumoniae | 29 |
| | | Proteus mirabilis | 15 |
| | | Pseudomonas aeruginosa | 9 |
| | Urine culture | Escherichia coli | 361 |
| | | Proteus mirabilis | 90 |
| | | Klebsiella pneumoniae | 84 |
| | | Enterococcus faecalis | 59 |
| | | Pseudomonas aeruginosa | 43 |
| | Other fluid culture | Escherichia coli | 13 |
| | | Staphylococcus aureus | 11 |
| | | Klebsiella pneumoniae | 5 |
| | | Streptococcus anginosus group | 4 |
| | | Enterococcus faecium | 2 |

2017, $N_{val} = 1218$ patient-infections from 2018, and $N_{test} = 1320$ patient-infections from 2019. In Table 3 we report the antibiotic susceptibility classifier performance in the test set and specifically show the best prediction model class (lasso, ridge, random forest, or gradient boosted tree), prevalence (fraction of patient-infections to which the antibiotic was listed as susceptible in the test set), average precision and the area under the receiver operating characteristics curve (AUROC) for each of the 12 antibiotic options[40]. Best model class refers to the model type that performed best in the validation set and was thus used in the final evaluation and downstream personalized antibiotic selection. The positive class prevalence (equivalent to normal antibiogram values and indicative of class balance) ranged from 0.23 (vancomycin) to 0.98 (vancomycin and meropenem). Average precision ranged from 0.46 to 0.99. AUROC ranged from 0.61 to 0.73.

In Supplementary Fig. 1 we show precision-recall curves for each classifier. In Supplementary Table 1 we show validation set performance of the best model for each model class and the chosen hyperparameters. In the Supplementary Table 2 we show number of features stratified by feature type. In Supplementary Table 3 we show the five most important features of each of the final 12 models. The cohort contained repeated observations per patient. In Supplementary Table 4 we show model performance

**Table 3 Antibiotic susceptibility classifier performance.**

| Antibiotic selection | Best model class | Prevalence | Average precision | AUROC |
|---|---|---|---|---|
| Vancomycin | Gradient Boosted Tree | 0.23 | 0.46 [0.40, 0.52] | 0.72 [0.68, 0.75] |
| Ampicillin | Gradient Boosted Tree | 0.43 | 0.54 [0.49, 0.58] | 0.62 [0.59, 0.65] |
| Cefazolin | Gradient Boosted Tree | 0.59 | 0.72 [0.68, 0.76] | 0.67 [0.64, 0.70] |
| Ciprofloxacin | Random Forest | 0.63 | 0.73 [0.70, 0.76] | 0.61 [0.58, 0.64] |
| Ceftriaxone | Gradient Boosted Tree | 0.66 | 0.79 [0.77, 0.82] | 0.69 [0.66, 0.72] |
| Cefepime | Random Forest | 0.80 | 0.87 [0.84, 0.89] | 0.65 [0.61, 0.69] |
| Vancomycin + Ceftriaxone | Gradient Boosted Tree | 0.81 | 0.87 [0.84, 0.89] | 0.67 [0.63, 0.71] |
| Meropenem | Gradient Boosted Tree | 0.82 | 0.90 [0.88, 0.92] | 0.69 [0.65, 0.72] |
| Pip-Tazo | Random Forest | 0.90 | 0.94 [0.92, 0.95] | 0.64 [0.59, 0.69] |
| Vancomycin + Pip-Tazo | Random Forest | 0.96 | 0.98 [0.97, 0.99] | 0.70 [0.62, 0.77] |
| Vancomycin + Cefepime | Random Forest | 0.97 | 0.98 [0.98, 0.99] | 0.70 [0.62, 0.78] |
| Vancomycin + Meropenem | Gradient Boosted Tree | 0.98 | 0.99 [0.99, 0.99] | 0.73 [0.65, 0.81] |

Pip-Tazo = piperacillin/tazobactam.

on the test set stratified by whether the patient had an observation in the combined training and validation set. In Supplementary Tables 5–16 we show model performance stratified by demographic data including age, race, sex, and ethnicity as well as insurance payer and whether the admission occurred at the Stanford or ValleyCare emergency departments.

**Personalized antibiogram guided antibiotic selection**. To estimate the potential clinical impact of prescribing guided by personalized antibiograms, we employ a linear programming based optimization procedure to simulate reassigning antibiotic selections to patients in our held out test set using personalized antibiogram estimates. We track the coverage rate (fraction of infections covered by the antibiotic selection) achieved by personalized antibigorams and compare to coverage rate achieved by clinician and random selections. The personalized antibiogram guided treatment selection and random selection are constrained such that the frequencies at which specific antibiotics are assigned to patients match the frequencies used by clinicians. We call these budget constraints. Without such a constraints, assignment of the broadest spectrum antibiotics possible would result in the highest coverage rate. We restrict this analysis to the subset of infections in our test set where one of the 12 antibiotic selections were administered. This results in a total of $N = 770$ observations. Antibiotic selection guided by personalized antibiograms in the Stanford data demonstrates a coverage rate of 85.9%; significantly better than random treatment selection (79.2% $p < 0.0001$) and comparable to clinician performance (84.3% $p = 0.11$). In Supplementary Fig. 2 we show the clinician coverage rate by antibiotic selection, and in Supplementary Table 17 we show the infections most frequently missed by clinicians.

**Promoting antibiotic stewardship with reduced broad-spectrum antibiotic use**. To assess the potential for reducing broad-spectrum antibiotics with personalized antibiograms, we sort antibiotic selections according to their antibiogram value (fraction of infections listed as susceptible) and repeat the linear programming based optimization simulations under revised constraints where the personalized antibiogram based treatment selection is forced to use fewer broad-spectrum (higher antibiogram value) antibiotics in place of more narrower-spectrum (lower antibiogram value) antibiotics. We track the point at which the coverage rate of the personalized antibiogram based antibiotic selections matched the real world coverage rate, and the point at which it matched the coverage rate of a random antibiotic selection. Results of these simulations are shown in Fig. 4.

Simulations for all pairs of broader and narrower antibiotic selections are shown in Supplementary Fig. 3.

We find that antibiotic selections guided by personalized antibiograms can achieve a coverage rate as good as the real world clinician prescribing rate while narrowing 69% of the vancomycin + piperacillin/tazobactam selections to piperacillin/tazobactam, 40% of piperacillin/tazobactam prescriptions to cefazolin, and 21% of ceftriaxone prescriptions to ampicillin.

**Replication on an external site**. We replicate our experiments on a held out dataset consisting of $N = 15{,}806$ uncomplicated urinary tract infections from 13,862 unique female patients who presented to Massachusetts General Hospital and Brigham & Women's Hospital in Boston between 2007 and 2016[47]. Mean age for the cohort is $34.0 \pm 10.9$ years, and 63.6% of infections were from Caucasian women. Further information on the exact cohort definition can be found in Kanjilal et al.[28].

We use the training and test set split used in Kanjilal et al. to train and evaluate personalized antibiogram models. The training set contained $N_{train} = 11{,}865$ patient-infections while the test contained $N_{test} = 3941$. We train personalized antibiogram models to predict the probability that four antibiotics administered to patients in this dataset with urinary tract infection would go on to be listed as susceptible for the patient's underlying infection. As in Kanjilal et al. the four antibiotic options are ciprofloxacin, levofloxacin, trimethoprim/sulfamethoxazole, and nitrofurantoin. Table 4 shows performance of our personalized antibiogram models listing the final model class, antibiogram value (mean susceptibility across infections), average precision and AUROC. In Supplementary Fig. 4 we show precision-recall curves for each classifier.

In Fig. 5 we show the results of the personalized antibiogram based treatment selection and show how the coverage rate changes as fewer broad-spectrum antibiotics are used across patients in the test set. Without changing the frequency at which antibiotics are used by actual clinicians in this dataset, personalized antibiogram based treatment selection achieve a coverage rate of 90.4%, significantly greater than both a random antibiotic selection (87.5% $p < 0.0001$) and real world clinician prescribing (88.1% $p < 0.0001$). Replications with varying linear programming budget parameters demonstrate that 93% of the total ciprofloxacin prescriptions could have been exchanged with nitrofurantoin without falling below the real world coverage rate. Similarly we show that 48% of the total ciprofloxacin and 62% of nitrofurantoin prescriptions could have been exchanged with trimethoprim/sulfamethoxazole.

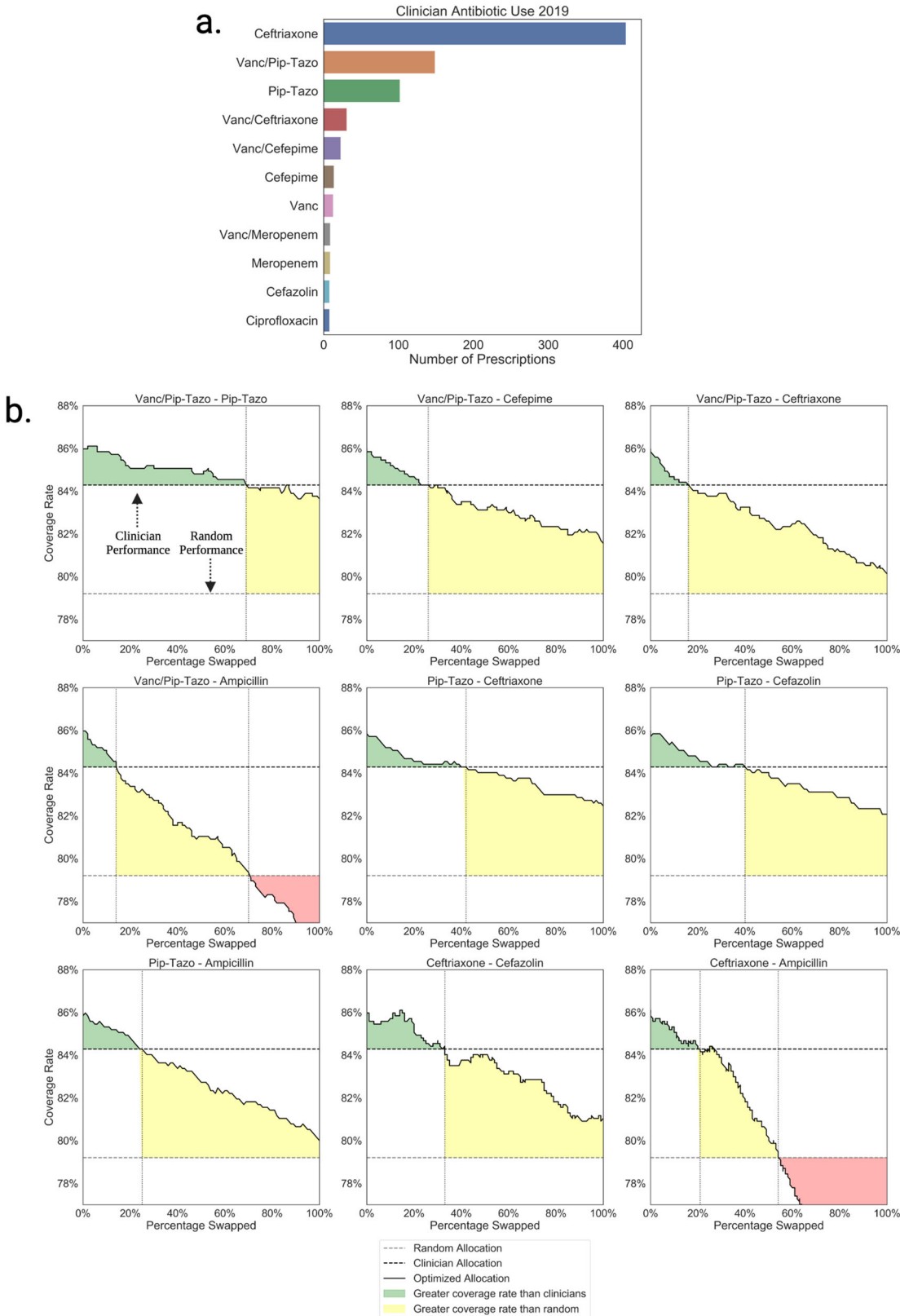

**Fig. 4 Personalized antibiogram coverage rate as a function of the antibiotic budget parameters. a** Held out Stanford test set of emergency hospital admissions treated with one of the twelve most common empiric IV/IM antibiotic selections. **b** Antibiotic sweep simulations showing the trade off between coverage rate and broad-spectrum antibiotic use when antibiotic selection is optimized with personalized antibiograms. For example, the plot titled "Vanc/Pip-Tazo - Pip-Tazo" illustrates the change in coverage rate as more historical prescriptions for Vancomycin + Piperacillin-Tazobactam (Vanc/Pip-Tazo) are exchanged for Pip-Tazo, prioritized by personalized antibiogram predictions. On the far left of the plot, starting with 0 change in the actual amount of antibiotics prescribed, the figure illustrates the green region where personalized antibiograms could potentially have achieved an even better antibiotic coverage rate than the actual prescriptions by clinicians (represented by the dark dashed line) with equal or fewer Vanc/Pip-Tazo.

**Table 4 Boston Model Performances.**

| Antibiotic selection | Best model class | Prevalence | Average precision | AUROC |
|---|---|---|---|---|
| Trime/Sulf | Gradient Boosted Tree | 0.80 | 0.85 [0.84, 0.87] | 0.60 [0.58, 0.62] |
| Nitrofurantoin | Gradient Boosted Tree | 0.89 | 0.91 [0.90, 0.92] | 0.57 [0.54, 0.61] |
| Ciprofloxacin | Lasso | 0.94 | 0.95 [0.95, 0.96] | 0.64 [0.60, 0.68] |
| Levofloxacin | Lasso | 0.94 | 0.96 [0.95, 0.96] | 0.64 [0.60, 0.67] |

Trime/Sulf = trimethoprim/sulfamethoxazole.

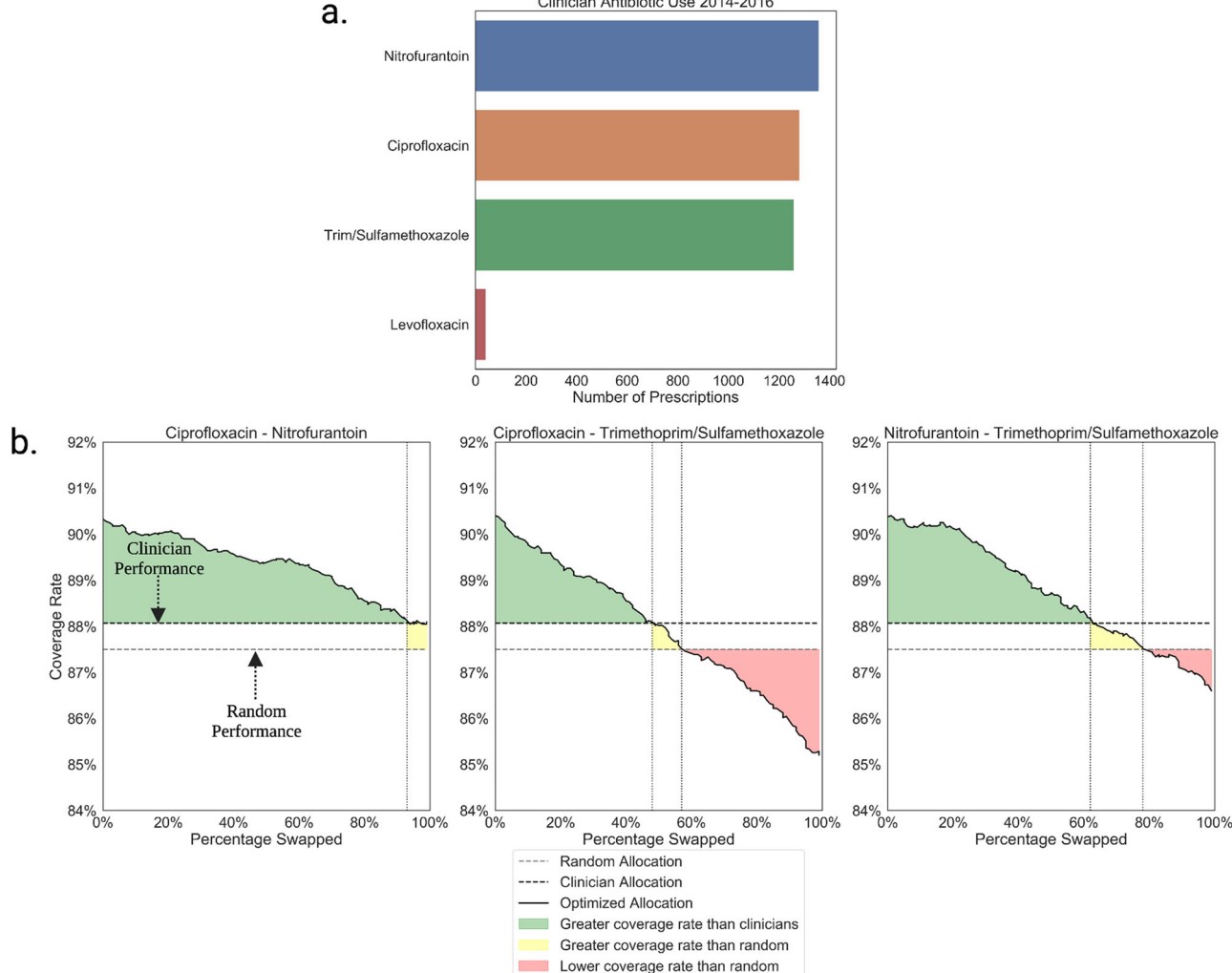

**Fig. 5 Personalized antibiogram coverage rate as a function of the antibiotic budget parameters in the Boston dataset. a** Held out Boston test set of uncomplicated urinary tract infection cases treated with one of the four antibiotics in the dataset. **b** Antibiotic sweep simulations showing the trade off between miss rate and broad-spectrum antibiotic use when antibiotic selection is optimized with personalized antibiograms.

**Sensitivity analysis**. We conduct a sensitivity analysis with the Stanford cohort to incorporate patients with negative microbial culture results into our model performance estimates. This is done by (1) constructing an electronic phenotype to flag patients with negative microbial cultures that lacked infection, (2) training twelve new personalized antibiogram models (sensitivity analysis models) that include patients with negative cultures, and (3) using inverse probability weighting model performance estimates[48].

In Table 5 we compare performance (as measured by AUROC) of the sensitivity analysis models with our original 12 personalized antibiogram models in the Stanford data. Specifically, we show performance of (1) the personalized antibiogram models trained and evaluated only on patients with positive microbial cultures, (2) performance of the sensitivity analysis models trained and evaluated on patients including those flagged by the electronic phenotype and (3) estimates of the sensitivity analysis model performances on the entire deployment population. Model performance increases when we incorporate patients with negative microbial cultures into the analysis. Model performance remains stable when incorporating the inverse probability weights into the estimation of AUROC.

## Discussion

Antibiotic susceptibility classifiers (personalized antibiograms) demonstrated modest to moderate discriminatory power in terms of AUROC, consistent with prior state-of-the-art literature.

**Table 5 Personalized antibiogram sensitivity analysis with and without inverse probability weights: Pip-Tazo = piperacillin/tazobactam.**

| Antibiotic selection | Original classifiers | Sensitivity analysis | |
| --- | --- | --- | --- |
| | AUROC | AUROC | AUROC$_{IPW}$ |
| Vancomycin | 0.72 [0.68, 0.75] | 0.74 [0.71, 0.76] | 0.75 [0.72, 0.77] |
| Ampicillin | 0.62 [0.59, 0.65] | 0.69 [0.66, 0.71] | 0.69 [0.66, 0.71] |
| Cefazolin | 0.67 [0.64, 0.70] | 0.71 [0.68, 0.73] | 0.70 [0.67, 0.73] |
| Ceftriaxone | 0.69 [0.66, 0.72] | 0.72 [0.69, 0.75] | 0.72 [0.69, 0.74] |
| Cefepime | 0.65 [0.61, 0.69] | 0.64 [0.60, 0.68] | 0.62 [0.58, 0.66] |
| Pip-Tazo | 0.64 [0.59, 0.69] | 0.65 [0.59, 0.70] | 0.62 [0.56, 0.68] |
| Ciprofloxacin | 0.61 [0.58, 0.64] | 0.64 [0.62, 0.68] | 0.64 [0.61, 0.67] |
| Meropenem | 0.69 [0.65, 0.72] | 0.71 [0.68, 0.74] | 0.70 [0.67, 0.74] |
| Vancomycin + Meropenem | 0.73 [0.65, 0.81] | 0.76 [0.67, 0.84] | 0.74 [0.65, 0.84] |
| Vancomycin + Pip-Tazo | 0.70 [0.62, 0.77] | 0.71 [0.63, 0.78] | 0.70 [0.62, 0.78] |
| Vancomycin + Cefepime | 0.70 [0.62, 0.78] | 0.68 [0.60, 0.77] | 0.67 [0.59, 0.76] |
| Vancomycin + Ceftriaxone | 0.67 [0.63, 0.71] | 0.71 [0.68, 0.75] | 0.70 [0.66, 0.74] |

AUROC alone however is not a good determinant of clinical utility[49]. Our optimization simulations demonstrate that even with modest AUROCs, antibiotic selection informed by personalized antibiograms can match or exceed clinician performance. Furthermore, antibiotic selection guided by personalized antibiograms achieved similar coverage rates to those seen in the real world with fewer broad-spectrum antibiotics, an ongoing and vital antibiotic stewardship challenge[50–52]. An example of which is the goal to reduce empiric vancomycin utilization. While vancomycin provides effective Gram positive coverage including methicillin resistant *Staphylococcus aureus* (MRSA), reducing unnecessary empiric vancomycin reduces antimicrobial resistance pressure and toxicities including ototoxicity and renal toxicity[52,53]. In 2019, the proportion of *Staphylococcus aureus* classifying as MRSA was 22% at Stanford, which likely explains the high frequency of vancomcyin + piperacillin/tazobactam empiric antibiotic use in the emergency department[54] in our test set. We demonstrated that personalized antibiogram guided antibiotic selection could reduce vancomycin + piperacillin/tazobactam use in favor of piperacillin/tazobactam monotherapy by 69% without falling below the coverage rate achieved by clinicians. Another antibiotic stewardship goal is to reduce use of fluoroquinolones for treatment of urinary tract infection[55]. Despite criteria outlined in the Infectious Disease Society of America guidelines, fluoroquinolone use across the United States remains high at 40.3% to treat cases of uncomplicated urinary tract infection[56]. In the Boston test set, flouroquinolone was used by clinicians across 33.6% of cases. We demonstrated that personalized antibiogram guided antibiotic selection could reduce ciprofloxacin use by up to 48% in favor of trimethoprim/sulfamethoxazole while maintaining the real world coverage rate. These results point toward the broader vision and impact of this work. The CDC identifies antibiotic stewardship as the most important intervention to combat the larger ecological costs of increasing microbial resistance[6]. In this study we empirically addressed the double-edged sword of ensuring patients receive immediate value in the clinical care they receive while minimizing overexposure of excessive broad-spectrum antibiotics. Our results indicate that personalized antibiograms could simultaneously promote antibiotic stewardship goals while ensuring, if not improving, patient safety.

Clinician performance however is not the only baseline worth benchmarking against. In additional experiments outlined in Supplementary Notes 1 and 2 we (1) compare the performance of our personalized antibiogram approach to a linear programming based antibiotic allocation that uses normal antibiogram values and (2) compare to a rule based algorithm on a subgroup of our cohort designed to mimic institutional guidelines for treating patients hospitalized with urinary tract infection.

The normal antibiogram approach, which also used our linear programming optimization procedure, outperformed the personalized antibiogram approach—though we note the comparison was not apples-to-apples as the normal antibiogram leveraged information about species identity to generate probability estimates of antibiotic susceptibility. At the time empiric antibiotic selection, the species is only suspected but not known. This analysis nevertheless demonstrates the utility of our linear programming based optimization framework in the advent of rapid diagnostic technology that can more quickly determine gram stain and species identity.

The personalized antibiogram approach outperformed the guideline based approach on the sub-population of patients in our cohort hospitalized with urinary tract infection. In the same subset, the guideline based approach outperformed clinician performance. We note however the comparison of guidelines to clinicians was also not apples-to-apples because in the subgroup of patients with urinary tract infection clinicians used a different set of antibiotics than what the guideline based approach allowed. This is likely due to the fact that at the point in time clinicians ordered empiric antibiotics, the exact syndrome, to which guidelines are tailored, was only suspected and not known. This further demonstrates the need for technology to enable guided antibiotic selection that models the uncertainty of the pathogen and syndrome—like our personalized antibiograms.

The success of clinical decision support relies heavily on whether it can be successfully implemented into existing clinical workflows. Systems need to be designed such that the "right information needed to make the right decision for the right patient at the right time." is conveyed appropriately[57]. We've designed personalized antibiograms to fit in with existing clinical workflows. When a patient presents with potential infection, clinicians first order a set of microbial cultures and provide empiric therapy based upon personal experience, clinical practice guidelines, or institutional standards[58,59]. Personalized antibiograms are designed to provide individualized antibiotic susceptibility predictions at the point in time clinicians are already seeking external support in the existing workflow. Prediction models could be embedded in electronic health records, such as through Epic cognitive compute environments, and in real time gather patient specific features, compute and display a susceptibility probability score for each antibiotic considered[21] A key direction for future research is thus to prospectively evaluate the

effect of personalized antibiogram based decision support on successful antibiotic selection.

Our linear programming based optimization procedure requires a specified fixed "budget" of each antibiotic choice across an entire population. In practice, patients present with infection sequentially one at a time. The optimization procedure simulated and presented here is thus not explicitly a decision algorithm intended for direct sequential use. Nevertheless, once solved with a particular set of constraints, the solution space is broken into regions that would allow antibiotic recommendations for future patients in a sequential manner.

We acknowledge limitations associated with the retrospective nature of our analysis, which leveraged electronic medical record data collected in an observational manner. Though the Stanford cohort is derived from a dataset representing a comprehensive primary/secondary healthcare system, we note that there still exists a possibility that patients included in this study had medical encounters elsewhere that went unobserved. We note however that our model performance estimates reflect any error that may be induced by this phenomenon.

Not all microbial cultures return positive, and those that are negative do not always indicate lack of bacterial infection. We attempted to address potential selection bias in our model performance estimates induced by this phenomenon in a sensitivity analysis estimating model performance on the full population (patients with both positive and negative microbial cultures) using inverse probability weighting. We acknowledge that this analysis relied on diagnosis codes to rule in possible infection, and requires assumptions about lack of unmeasured confounders. Nevertheless, the results of this analysis reassuringly revealed no considerable change in performance between the two settings, within the constraints of the observed data.

We acknowledge limitations associated with the use of microbial cultures as a gold standard. Positive microbial cultures do not always indicate a true clinical infection. *Coagulase-Negative Staphylococci* is a common contaminant in blood cultures which we handled by excluding them from our set of infections. *Enterococcus* is sometimes considered a colonizer in urine cultures, especially when patients lack symptoms. To address this limitation, we only included in our analysis patient encounters where prescribers clinically determined hospitalization and empiric antibiotic treatment was needed based on patient presentation. We also acknowledge that microbial culture results can yield imperfect labels due to the presence of ESCHAPPM organisms[60]. In-vitro these organisms may appear susceptible to cephalosporins but in-vivo can induce resistance.

By the nature of differing antibiotic susceptibility patterns across institutions, individual models are likely not robust to distributional shifts. We expect model performance could degrade if applied to different patient populations, though many of the risk factors for drug resistant organism infection are likely transferable to different settings (e.g., previous hospitalization and exposure to antibiotics and resistant infections). Specifically, we would expect our models trained on Stanford data to perform worse on Boston data and vice-versa. Future work could characterize the degree to which model performance decays across sites and the utility of retraining. Here however we demonstrate that the process of training personalized antibiogram models and using them to inform antibiotic selection is generalizable by repeating our analysis on held out data from an external site. We would also expect model performance to naturally degrade over time, which is why we separated our training and test sets by time to offer a more realistic appraisal of their potential future performance.

In the Stanford cohort, antibiotic selection guided by personalized antibiograms achieved coverage rates similar to clinicians, while in the Boston dataset coverage rates increased. It should be noted that in the Boston dataset, clinician performance is closer to random chance. In the Stanford cohort the diversity of antibiotics commonly ordered was much greater as there was more variation in the types of cases presented as opposed to the more narrow uncomplicated urinary tract infection case in the Boston cohort. Analysis results can be sensitive to the details of any cohort definition. Nevertheless, personalized antibiograms demonstrate promise in multiple settings.

Lastly, not all infections present the same risk to patients if treated with an inappropriate antibiotic regimen. Future work may benefit from leveraging the idea that some infections are more critical to treat than others.

## Conclusion

Machine learning classifiers trained using electronic health record data can predict antibiotic susceptibility for patients with positive microbial cultures. Antibiotic selection policies guided by personalized antibiograms could maintain or improve infection coverage rates while using fewer broad-spectrum antibiotics than are seen in real-world practice. Machine learning driven antibiotic selection could improve antibiotic stewardship without sacrificing, and potentially even improving, patient safety.

### Data availability

The Stanford data is made available through STARR, STAnford medicine Research data Repository[30]. The data can be accessed for research purposes after Institutional Review Board approval via the Stanford Research Informatics Center. The Boston cohort has been made available through Physionet for credentialed users who sign the specified data usage agreement[32]. The source data for Figs. 4 and 5 have been provided alongside this article as Supplementary Data 1 and Supplementary Data 2.

### Code availability

Code developed for this study has been made available on GitHub and deposited in a DOI minting repository using Zenodo[61].

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

## Acknowledgements

This research used data provided by STARR, STAnford medicine Research data Repository," a clinical data warehouse containing de-identified Epic data from Stanford Health Care (SHC), the University Healthcare Alliance (UHA) and Packard Children's Health Alliance (PCHA) clinics and other auxiliary data from Hospital applications such as radiology PACS. The STARR platform is developed and operated by the Stanford Medicine Research IT team and is made possible by Stanford School of Medicine Research Office. Insight into the operations of Stanford's microbiology lab was provided by Nancy Watz and Dr. Niaz Banaei. Rich Medford and Kojo Osei contributed to project ideation. The content is solely the responsibility of the authors and does not necessarily represent the official views of the NIH or Stanford Health Care. Figures 1, 2, and 3 were created with BioRender.com.

## Author contributions

C.K.C. and J.H.C. conceived of the study. C.K.C. processed the data, trained the machine learning models, conceived of the linear programming optimization formulation, and drafted the initial paper. L.S. and J.H.C. critically revised the paper. A.r.C. performed quality control on the data. M.N. performed analysis on the Boston cohort. M.B. provided statistical insight. A.m.C., A.r.C., S.D., J.H.C., and L.S. provided clinical insight. All authors contributed to the study and analysis design and reviewed the final paper.

## Competing interests

The authors declare the following competing interests. C.K.C. receives consultation payment from Fountain Therapeutics, Inc. C.K.C. was an intern at Verily Life Sciences while drafting the paper. J.H.C. co-founded of Reaction Explorer LLC that develops and licenses organic chemistry education software and receives consultation payment from the National Institute of Drug Abuse Clinical Trials Network, Tuolc Inc., Roche Inc., and Younker Hyde MacFarlane PLLC. M.N. receives consultation payment from Vida Health. All other authors declare no competing interests.
