## [Peer Review File · Communications Medicine]

Reviewers' comments:

Reviewer #1 (Remarks to the Author):

This is a nicely written and interesting paper exploring precision empiric antibiotic prescribing supported by personalized antibiogram prediction models. The paper has a valuable contribution to the field of studies and proper research methodology. The discussion is clear and credible. The topic is up-to-date and relevant to the field of studies.

However, I have some comments that need to be addressed before this article can be accepted for publication in Communications Medicine journal.

- As there is a similar work [25], I think that a more extensive reference should be made to the introduction and / or discussion for this study and what more present study offers compared to the previous one.

[25] Conor K Corbin et al. "Personalized Antibiograms: Machine Learning for Precision Selection of Empiric Antibiotics". In: AMIA Summits on Translational Science Proceedings 2020 (2020), p. 108. <https://www.ncbi.nlm.nih.gov/pmc/articles/PMC7233062/>

- Is there any class imbalance? If yes, the authors should report it and describe the method that they address it?

- The authors should justify why they selected these specific machine learning classifiers, and I think that there must be at least a short description of each classifier.

- If too many features are used in the model, the risk of overfitting is very high. How do the authors address this issue?

- The authors should report if any features were standardized.

- The authors should justify why they split the cohort by year into training (2009-2017), validation (2018), and test (2019)? Did the authors take into account the evolution of antibiotic resistance over time?

- Is it possible to add some indicative figures of precision-recall curves for the test set to the manuscript or Supplementary Materials?

- Are there any missing values in this study except the missing labels that authors reported in the Labelling Infections for Personalized Antibiogram Models section? If yes, the authors should report it and describe the method that they impute them.

Reviewer #2 (Remarks to the Author):

This is a thoughtfully executed and generally well written piece of work. There is a lot of jargon which I find confusing. I am not sure whether these are American terms in general use, or statistical. I mention a few below but this may be my ignorance.

It adds to the body of data showing that machine learning has the potential to perform as well as medical staff in the selection of empirical antibiotics - or perhaps better with reductions in broad spectrum antibiotic use.

I am not a statistician and assume other reviewers with expertise in this field are taking a detailed look at the techniques used.

From a clinical perspective the scale of this work (assessing prediction of sensitivity to 12 different abx regimens) is novel as is the broad cohort.

Section "Cohort Description"

"repeated observations per patient to mimic the real world scenario" - don't understand this. Do you mean some patients had multiple admissions and had several antibiotic initiations?

Section "Personalised Antibigrams"

It would be useful to know more about the features used. For example are patients likely to be seen by other healthcare providers to whom the researchers would not have access? Does "medications" include those prescribed in primary care, arguably a greater driver of resistance. Or does this dataset represent a comprehensive primary/secondary care health system covering an entire geography?

I was not clear from table 3 whether the "Best Model Class" was the one used in personalised antibiogram selection. I assume so.

Section "Personalised Antibiogram guided antibiotic selection"

As a clinician I did not understand what "linear programme based optimisation procedure" was. I think it would be useful to have a visual representation of how selection happened. I assume each of the classifiers was applied in turn.

"Replication on an external site"

This is a key part of any such project - whether it achieves similar results in another cohort. It is unfortunate that it is just 4 antibiotics and patients with UTIs as this restricted clinical setting is not directly comparable with the broader Stanford patient dataset. But I assume this was a well characterised database to which the researchers had easy access. Did you try to apply the systems trained on Stanford data to the Massachusetts data? A lot of literature assumes this is needed but is it really? Is a system trained in one city in the USA actually fine for another? It would be useful to know whether performance was improved by retraining or whether the gains are marginal.

"Discussion"

Good points on AUROC not necessarily clearly relating to clinical utility - certainly we found similar.

I think an absolutely key point is how patients with negative microbiology, or not having infection, skew the performance of these models. The only way to know for sure would be a real-life prospective assessment but these authors go some way to assessing this by modelling it. This is currently squeezed into the discussion and supplementary material ("sensitivity analysis"). I think this is a key part and deserves to have a proper section in the results.

I don't understand the sentence "In the Stanford cohort the decision space... is much wider". Do you

mean there are no written empirical guidelines in that hospital? Or that clinicians don't follow them? Or they suggest multiple options? Or that because that data had lots of different conditions abx use was more diverse than the UTI dataset?

Reviewer #3 (Remarks to the Author):

Thank you for the opportunity to review this manuscript exploring the development of personalised antibiograms for antibiotic selection using machine learning. This is a timely and interesting manuscript, that overall is well written.

The authors set out to explore the following objectives:

1. Train and evaluate personalized antibiograms using routinely available EHR data.
2. Evaluate the performance of antibiotic selection informed by antibiograms compared for clinician choice.
3. Systematically evaluate the trade-off in performance when less broad-spectrum antibiotics are selected.

They demonstrate reasonable model performance and indicate that personalised antibiograms have the potential to support narrowing of the spectrum of antimicrobial prescribing against current clinician practice and known susceptibility profiles.

I believe that the manuscript may benefit from further methodological description. The limitations of using retrospective EHR data for this type of algorithm development could also be further explored.

In addition, please see some further thoughts below.

From a clinician's perspective, my thoughts are as follows:

1. The main question is what difference would this make c.f. gold standard (and what is the gold standard)? Comparing to clinician prescribing practice is not a good marker of performance. Comparing to antibiogram is useful as is potentially a more objective standard (particularly for sterile site infections, such as blood stream infection).
2. For urinary tract infection (UTI) I would be interested to understand how contamination / asymptomatic bacteraemia were controlled for in the data selection? Enterococcus is often felt to be a contaminant from the perineum, particularly in female patients – was this considered / do you think that your high rate of Enterococci in urinary samples are true UTI's (i.e. should antibiotics have been prescribed for this at all)? Were WBC counts in urine / epithelial cell presence considered?
3. Abx prescribing practices described here are very different in terms of empiric therapy in our local ED's. Is there a rationale for empiric use of vancomycin & tazocin compared to just ceftriaxone in patients with GPC in this region (e.g. high rates of MRSA)? More context would be useful for readers outside of the US.
4. In a similar fashion, it would be good to understand the local resistance patterns of organisms commonly observed here – is there a high rate of ESBL / MRSA observed in isolates?
5. It would also be good to know what guidelines recommend and how clinicians perform compared to local guidelines – for example, is it that clinicians are prescribing broad due to their own concerns or is use of vancomycin + pip-tazo empiric recommendation for suspected blood stream infection – in which case this would further compound your comparison of current clinician prescribing (as you

are actually comparing to local guidelines). If the latter is the case, then the question is whether simply changing the guidelines would further improve clinician performance (above that of your antibiograms)?

6. Was the same microbiology lab / SOP used for ID and sensitivities of all organisms? What happened if different laboratories use different methods of identification / susceptibility testing / susceptibility testing in the laboratory changed over the period of time analysed (e.g. CLSI increase / reduce the breakpoint for an organism)?

7. What about risk of selecting out AMR? Would be good to predict the ecological cost that these abx may have and risk stratify?

8. How did you handle ESBL / ampC producers where the in vitro phenotype might be susceptible, but in-vivo we know the organism is likely to be resistant / derepress (although I note that ESCHAPM organism do not feature heavily in your selected cultures).

9. What are the indications for wide spread quinolone use in uncomplicated UTI? This is not something that would normally be considered unless very high rates of ESBL or upper urinary tract infection is suspected / prostate issues? I think that overall, further justification of the agents selected would be beneficial for non-native readers.

From a ML development perspective:

10. In terms of the limitations of predictive models, this study is hampered by similar challenges highlighted in the introduction. Namely the reliance on retrospective data and using culture as a gold standard. The use of ICD-10 codes is not the best source for diagnosis of infection and needs to be acknowledged as a limitation.

11. In terms of baseline data how was class imbalance / missingness of data dealt with?

12. It would be interesting to consider what would happen in terms of the predictive value of your algorithms in the face of different prevalence rates of resistance. For example, if the algorithm were deployed in a region with very low / high ESBL or MRSA rates, would this make a large difference? If so would it be a potential consideration for what prescriptions / conditions to target in the future?

Reviewer Comments and Revisions

Referee expertise:

Referee #1: ML, including for antibiotic prescribing

Referee #2: Clinical infectious diseases, ML-guided antibiotic prescribing

Referee #3: Clinical infectious diseases, ML-guided antibiotic prescribing

Legend

Reviewer comments

Author responses

Reviewer #1 (Remarks to the Author):

This is a nicely written and interesting paper exploring precision empiric antibiotic prescribing supported by personalized antibiogram prediction models. The paper has a valuable contribution to the field of studies and proper research methodology. The discussion is clear and credible. The topic is up-to-date and relevant to the field of studies.

However, I have some comments that need to be addressed before this article can be accepted for publication in Communications Medicine journal.

- As there is a similar work [25], I think that a more extensive reference should be made to the introduction and / or discussion for this study and what more present study offers compared to the previous one.

[25] Conor K Corbin et al. "Personalized Antibiograms: Machine Learning for Precision Selection of Empiric Antibiotics". In: AMIA Summits on Translational Science Proceedings 2020 (2020), p. 108. <https://www.ncbi.nlm.nih.gov/pmc/articles/PMC7233062/>

Thank you, we agree further clarification of the difference between this and prior work would make our submission stronger. Our prior work focused on training and evaluating personalized antibiogram models that were species specific (unlike species independent models in this manuscript) and did not use model predictions to simulate antibiotic prescriptions. Our linear programming formulation is distinct to our current manuscript, as is our analysis estimating the trade-off in patient coverage rates as more narrow-spectrum antibiotics are used over a population. We also conducted a sensitivity analysis that estimates performance of our models on the union of patients with positive and negative microbial cultures that is distinct to this work.

We have added a paragraph at the end of our introduction section to clarify how our work submitted to nature communications medicine substantially extends our work submitted to the AMIA Informatics Summit.

- Is there any class imbalance? If yes, the authors should report it and describe the method that they address it?

We thank the reviewer for this suggestion. There indeed were varying levels of class imbalance for each of the prediction tasks. Table's 3 and 4 show the prevalence of the positive class for each task, and we have added a sentence in the second paragraph of the results section titled "Personalized Antibigrams" to better call attention to this.

Because we aren't concerned with model performance at the standard cut-off threshold of 0.5 and would rather not sacrifice model calibration in our test set we did not re-weigh or re-sample our training set observations as some often do to handle class imbalance. We do however report AUROC which has a standard random chance baseline of 0.5 with class balance of any kind alongside average precision and positive class prevalence for each task to give the reader insight into model performance in light of existing class imbalance.

We have added a justification for not re-weighing or re-sampling our training data to the first paragraph in the methods section titled 'Training and Model Selection Procedure'.

- The authors should justify why they selected these specific machine learning classifiers, and I think that there must be at least a short description of each classifier.

Thank you, the four model classes were chosen specifically so that we could sweep across different regions of the bias/variance trade-off and select a model (and hyperparameter setting) that best straddled the line between under and overfitting the data.

L1 and L2 logistic regressions are higher bias, less flexible model classes. The regularization terms added to the loss function reduce risk to overfitting when the number of features is large. These model classes however have a higher risk of underfitting the data (as compared to the more flexible random forest and gradient boosted tree models) as they model the data under assumptions of linearity.

The tree based models (random forest and gradient boosted trees) are more flexible (higher variance) than the logistic regressions as they allow for non-linear interactions. These model classes have a higher risk of overfitting the data (as compared to the logistic regressions) due to this flexibility.

We have added a justification and short description of each classifier type including a reference to Elements of Statistical Learning in the second paragraph of the methods section titled "Training and Model Selection Procedure".

- If too many features are used in the model, the risk of overfitting is very high. How do the authors address this issue?

We thank the reviewer for bringing up this important issue.

Our choice of model classes (logistic regressions with various regularization terms and tree based models) was done specifically to overcome challenges involved with overfitting the data — particularly when the number of features is large. We performed a thorough model selection procedure to select an optimal model class and set of hyperparameters to best straddle the line between over and underfitting the data. We used a validation set to choose the optimal model class. That is, we used a held out dataset (distinct from our final test set) to choose a classifier type + hyperparameter configuration that maximized generalization performance.

Finally, we used an independent test set to assess generalization performance of the final model to ensure we were not overfitting to our validation set.

We have expanded on our model selection procedure (second paragraph of the methods section titled “Training and Model Selection Procedure”) to make this more clear.

- The authors should report if any features were standardized.

Thank you, we agree this is important to clarify. We did not standardized our feature matrix, and instead left them as counts as is common for bag of words feature representations. In our bag of words representation, features were tokenized and the value associated with each feature was the number of times that feature was present in the patient’s medical record over a fixed look back window from the prediction time (index time). Numerical values were tokenized by binning them into deciles, where bin thresholds were built up using data only from the training set to prevent leakage.

We have added to the first and second paragraph of the methods section titled “Feature Engineering” to clarify the above.

- The authors should justify why they split the cohort by year into training (2009-2017), validation (2018), and test (2019)? Did the authors take into account the evolution of antibiotic resistance over time?

We thank the reviewer for bringing up this important discussion point. Clinical machine learning models are susceptible to performance decay due to dataset-drift — changes in the data generating process (ex prevalence of an outcome, changes in medical practice) that occur over time. We split our training, validation, and test sets by time to take into account model performance decay due to this phenomenon. We note that this can often lead to more conservative estimates of model performance as compared to random train/validation/test splits.

We have added a justification in the first paragraph of the methods section titled “Training and Model Selection Procedure” and have added a citation to the work of Jung et al (“Implications of non-stationarity on predictive modeling using EHRs”) that emphasizes the importance of a time split over a random split for clinical predictive models.

- Is it possible to add some indicative figures of precision-recall curves for the test set to the manuscript or Supplementary Materials?

Thank you, we agree the addition of precision-recall-curves would improve our submission. We have added precision-recall curves for each of the twelve Stanford models and each of the four Bostom models to the supplementary material section "Precision Recall Curves" Figures 1 and 2.

- Are there any missing values in this study except the missing labels that authors reported in the Labelling Infections for Personalized Antibigram Models section? If yes, the authors should report it and describe the method that they impute them.

We thank the reviewer for allowing us to clarify this important point. Our bag of words feature representation allowed us to implicitly encode missing values into our feature matrix — which is especially beneficial when the fact that a feature is missing (ex: a lab test wasn't ordered) may be informative. Because missingness is implicitly encoded (as a zero), we did not need to impute any data in our feature matrix.

We have added a sentence to the end of the first paragraph in the methods section titled "Feature Engineering" to make this more clear.

Reviewer #2 (Remarks to the Author):

This is a thoughtfully executed and general well written piece of work. There is a lot of jargon which I find confusing. I am not sure whether these are American terms in general use, or statistical. I mention a few below but this may be my ignorance.

It adds to the body of data showing that machine learning has the potential to perform as well as medical staff in the selection of empirical antibiotics - or perhaps better with reductions in broad spectrum antibiotic use.

I am not a statistician and assume other reviewers with expertise in this field are taking a detailed look at the techniques used.

From a clinical perspective the scale of this work (assessing prediction of sensitivity to 12 different abx regimes) is novel as is the broad cohort.

Section "Cohort Description"

"repeated observations per patient to mimic the real world scenario" - don't understand this. Do you mean some patients had multiple admissions and had several antibiotic initiations?

We thank the reviewer for helping us clarify. Yes, here we intend to convey and justify how we have constructed our dataset (and specifically our test set) to best represent conditions the

models would encounter in a hypothetical deployment scenario. In deployment models would encounter patients with new infections who had previously been admitted to the same institution with prior infections. Thus when constructing our test set, we did not remove observations from patients who had prior infections in the training set.

We note that due to this design choice the resulting model performance estimates may be overestimates of performance on entirely new populations from unseen patients, which is why in the supplemental materials table 4 we show performance estimates on the subset of patients in our test set who had not been seen during training — even though these estimates likely underestimate performance in a the deployment population.

We have changed the wording and expanded upon this design choice in the method section titled “Cohort Description”.

Section "Personalised Antibigrams"

It would be useful to know more about the features used. For example are patients likely to be seen by other healthcare providers to whom the researchers would not have access? Does "medications" include those prescribed in primary care, arguably a greater driver of resistance. Or does this dataset represent a comprehensive primary/secondary care health system covering an entire geogrpaphy?

Thank you, the Stanford dataset is representative of a comprehensive primary/secondary healthcare system. We restrict our cohort to only include patient infections that required hospital admission, but features used in the models were not restricted to data collected from hospital encounters. Medications prescribed in primary care among other ambulatory settings were used to construct the feature matrix.

We have added clarifying text to the first paragraph of the methods section titled “Data Source” to make this more clear.

We have further added a sentence in the first paragraph of the method section titled “Feature Engineering” to make clear that we do not restrict features to data collected during hospital admissions.

And finally, while the dataset is a comprehensive primary/secondary system it is still possible that patients in our cohort had some medical encounters that would not appear in our dataset. We have added this limitation to our discussion section in the paragraph starting with “We acknowledge limitations associated with the retrospective...”

I was not clear from table 3 whether the "Best Model Class" was the one used in personalised antibiogram selection. I assume so.

We thank the reviewer for helping us clarify. Yes, in Table 3 “Best Model Class” refers to the type of machine learning classifier that performed best on the validation set. This classifier type was then trained on the combined training and validation sets for final evaluation on the test set, and used for the personalized antibiogram selection.

We have added a clarifying sentence in the paragraph starting with “The dataset was split by time into training...” in the results section titled “Personalized Antibiograms”

Section "Personalised Antibiogram guided antibiotic selection"

As a clinician I did not understand what "linear programme based optimisation procedure" was. I think it would be useful to have a visual representation of how selection happened. I assume each of the classifiers was applied in turn.

Thank you, we agree that our submission could benefit from a visual representation of the linear programming based optimization procedure. We have created a new figure (Figure 2 in revised manuscript) that demonstrates this procedure on a toy example.

"Replication on an external site"

This is a key part of any such project - whether it achieves similar results in another cohort. It is unfortunate that it is just 4 antibiotics and patients with UTIs as this restricted clinical setting is not directly comparable with the broader Stanford patient dataset. But I assume this was a well characterised database to which the researchers had easy access. Did you try to apply the systems trained on Stanford data to the Massachusetts data? A lot of literature assumes this is needed but is it really? Is a system trained in one city in the USA actually fine for another? It would be useful to know whether performance was improved by retraining or whether the gains are marginal.

Thank you, we agree this is an important discussion point. We would expect model performance to decay when tested on a different site due to changes in resistance patterns and variations in clinical practice. We focused on the Massachusetts data as it is one of the only applicable publicly available datasets for this question. With some variation in the underlying data components, we applied our overall training and testing process on the Massachusetts data to confirm reproducibility of the system process. Just as each institution should have its own local antibiogram, we would not expect personalized antibiogram models trained at one site to be directly usable at another without retraining.

We agree that this point deserves attention in our manuscript, and thus have added text to the paragraph in the discussion section starting with “By the nature of differing antibiotic susceptibility patterns...”

"Discussion"

Good points on AUROC not necessarily clearly relating to clinical utility - certainly we found similar.

I think an absolutely key point is how patients with negative microbiology, or not having infection, skew the performance of these models. The only way to know for sure would be a real-life prospective assessment but these authors go some way to assessing this by modelling it. This is currently squeezed into the discussion and supplementary material ("sensitivity analysis"). I think this is a key part and deserves to have a proper section in the results.

We thank the reviewer for this suggestion. We agree that our sensitivity analysis is a key part of our submission, and have thus substantially re-worked the portions of the manuscript and supplementary material related to it to give it more of a spotlight in the main text. Specifically, we have added entire sections titled "Sensitivity Analysis" in both the results and methods section in the main text.

I don't understand the sentence "In the Stanford cohort the decision space... is much wider". Do you mean there are no written empirical guidelines in that hospital? Or that clinicians don't follow them? Or they suggest multiple options? Or that because that data had lots of different conditions abx use was more diverse than the UTI dataset?

We thank the reviewer for this comment and for providing language that will help us make this point more clear to readers. By stating that the decision space is wider, we mean that antibiotic use was more diverse in the Stanford dataset because it contained observations for many more conditions (not just UTI).

We have reworked the language in this discussion paragraph starting with "In the Stanford cohort, antibiotic selection guided by..." to make this point more clear.

Reviewer #3 (Remarks to the Author):

Thank you for the opportunity to review this manuscript exploring the development of personalised antibiograms for antibiotic selection using machine learning. This is a timely and interesting manuscript, that overall is well written.

The authors set out to explore the following objectives:

1. Train and evaluate personalized antibiograms using routinely available EHR data.
2. Evaluate the performance of antibiotic selection informed by antibiograms compared for clinician choice.
3. Systematically evaluate the trade-off in performance when less broad-spectrum antibiotics are selected.

They demonstrate reasonable model performance and indicate that personalised antibiograms have the potential to support narrowing of the spectrum of antimicrobial prescribing against current clinician practice and known susceptibility profiles.

I believe that the manuscript may benefit from further methodological description. The limitations of using retrospective EHR data for this type of algorithm development could also be further explored.

In addition, please see some further thoughts below.

From a clinician's perspective, my thoughts are as follows:

1. The main question is what difference would this make c.f. gold standard (and what is the gold standard)? Comparing to clinician prescribing practice is not a good marker of performance. Comparing to antibiogram is useful as is potentially a more objective standard (particularly for sterile site infections, such as blood stream infection).

Thank you, the primary reason for not originally comparing directly to the normal antibiogram is because using species specific probabilities would give this baseline an unfair advantage — due to the fact that when empiric antibiotics are ordered, the species is only suspected and not known. While we note this isn't a fair apples-to-apples comparison, we conducted additional experiments to address this comment as we believe this could demonstrate how well a clinical decision support system *could* do if it knew gram stain or species identity at the time of the recommendation — something possible in the future with advancing rapid diagnostic technology.

We conducted two additional experiments, the results of which demonstrate the promise of our linear programming based optimization tool when using gram stain and species specific susceptibility probabilities. Due to space constraints in the main portion of the text, we mention these experiments in the discussion section in the paragraph that starts with “Clinician performance however is not the only...” and detail the full experiment and results in supplementary material under the section titled “Antibiogram Based Prescribing”

2. For urinary tract infection (UTI) I would be interested to understand how contamination / asymptomatic bacteriuria were controlled for in the data selection? Enterococcus is often felt to be a contaminant from the perineum, particularly in female patients – was this considered / do you think that your high rate of Enterococci in urinary samples are true UTI's (i.e. should antibiotics have been prescribed for this at all)? Were WBC counts in urine / epithelial cell presence considered?

We thank the reviewer for bringing up this excellent discussion point. We acknowledge the limitation for false positive (and false negative) microbial cultures. Indeed we do have institutional guidelines that recommend considering *Enterococcus* a colonizer when the patient is asymptomatic. As an attempt to isolate clinical infections, we only include in our analysis patient encounters that prescribers clinically determined warranted hospitalization and empiric antibiotic treatment based on patient presentation. We have added this limitation to the discussion section in the paragraph that starts with “We acknowledge limitations associated with the use of microbial cultures...”

3. Abx prescribing practices described here are very different in terms of empiric therapy in our local ED's. Is there a rationale for empiric use of vancomycin & tazocin compared to just ceftriaxone in patients with GPC in this region (e.g. high rates of MRSA)? More context would be useful for readers outside of the US.

We thank the reviewer for this excellent comment. Yes, based on our local and national guidelines, the prevalence of MRSA is high enough (22% of all *Staphylococcus aureus*) in our hospitalized patients that empiric coverage with vancomycin and tazocin was often suggested during the periods studied. Note we only include patients who were sick enough for hospital admission in our analysis. This reflects an important motivation for our work — just because broad-spectrum antibiotic use may be warranted in local and national guidelines does not mean they are always needed. We hope that this work can contribute to this commentary, demonstrating that using more personalized information for precisely prescribing the minimal but sufficient antibiotics necessary is possible.

We have added a sentence in the first paragraph of the discussion section noting the prevalence of MRSA among all *Staphylococcus aureus* isolates in 2019 at Stanford and referencing the 2019 Stanford antibiogram.

4. In a similar fashion, it would be good to understand the local resistance patterns of organisms commonly observed here – is there a high rate of ESBL / MRSA observed in isolates?

Thank you. For maximal transparency, we reference the 2019 version of the Stanford antibiogram in the first paragraph of the discussion section (done to address prior comment) and additionally we added a section to the supplementary materials that provides context and links directly to it. The antibiogram shows the relative prevalence of MRSA vs MSSA and susceptibility patterns of various gram negative organisms including *E. Coli*. This should provide context to readers (e.g., in 2019, 22% of *Staphylococcus aureus* species were MRSA and 12% of *E. coli* were ceftriaxone resistant).

This was added to the supplementary materials section “Stanford Bugs and Drugs”.

5. It would also be good to know what guidelines recommend and how clinicians perform compared to local guidelines – for example, is it that clinicians are prescribing broad due to their own concerns or is use of vancomycin + pip-tazo empiric recommendation for suspected blood stream infection – in which case this would further compound your comparison of current clinician prescribing (as you are actually comparing to local guidelines). If the latter is the case, then the question is whether simply changing the guidelines would further improve clinician performance (above that of your antibiograms)?

We thank the reviewer for this comment. We agree that comparison to local clinical practice guidelines would provide additional context to this study. We have designed and executed an additional subgroup analysis experiment in an attempt to deconvolve clinicians prescribing patterns from patterns recommended by local guidelines. In this additional analysis, we find that

the personalized antibiogram approach outperformed the guideline approach given the same antibiotic “budget.” The guideline approach outperformed clinicians, though we note this was not a fair apples-to-apples comparison because clinicians used a different “budget” of antibiotics than what the reconstructed guideline based algorithm suggested.

Additionally, one key challenge is that many of our local guidelines are pathogen or syndrome specific. In a similar vein to our remarks about the comparison to an institutional antibiogram, at the time an empiric antibiotic selection is made the causal pathogen or syndrome is only suspected and not known. This further motivates the development of technology that can provide decision support without requiring concrete knowledge of the pathogen and syndrome, as is the case with our personalized antibiogram prescribing procedure.

We mention the additional subgroup analysis in the discussion paragraph starting with “Clinician performance however is not the only...” and point the reader to the section in the supplementary material titled “Benchmarking Against Clinical Practice Guidelines” for the analysis details and results.

6. Was the same microbiology lab / SOP used for ID and sensitivities of all organisms? What happened if different laboratories use different methods of identification / susceptibility testing / susceptibility testing in the laboratory changed over the period of time analysed (e.g. CLSI increase / reduce the breakpoint for an organism)?

We thank the reviewer for bringing up this important discussion point. The ValleyCare and Stanford emergency departments have different microbiology labs, however we contacted their respective laboratory specialists and the response was that the laboratory follows national standards for susceptibility testing in accordance with the CLSI. Changes in CLSI standards are an example of dataset shift that can happen over time, which is why it was important that we split our training, validation and test set by time to effectively model any decay in performance due to these changes in the data-distribution.

We have added text to the second paragraph of the method’s section titled “Data Source” to make it clear that antibiotic susceptibility data from the ValleyCare and Stanford emergency departments were collected from two separate microbiology laboratories.

7. What about risk of selecting out AMR? Would be good to predict the ecological cost that these abx may have and risk stratify?

Yes, this is exactly the important broader implication we envision this work moving towards. There exists a double edged sword, where on one end we need to provide immediate value in clinical care for patients in front of us. On the other end, we need to minimize risk of side effects and overexposure to excessively broad-spectrum antibiotics to curb growing resistance. Predicting the ecological cost of antibiotic exposure would require a different kind of study, but is the larger motivation for the study we have conducted here — especially as the CDC has

identified improved stewardship of antibiotics as the most important intervention to combat the larger ecological impacts of increasing antimicrobial resistance.

We have added more context to this narrative in the first paragraph of our discussion section.

8. How did you handle ESBL / ampC producers where the in vitro phenotype might be susceptible, but in-vivo we know the organism is likely to be resistant / derepress (although I note that ESCHAPM organism do not feature heavily in your selected cultures).

This is a critical insight to discuss. In fact, our labs include comments in culture results where in vitro the organism may appear susceptible but in-vivo can induce resistance. We have added a section to our limitations that discusses this phenomenon and its implications on the interpretation of our results. As the reviewer notes, ESCHAPM organisms overall represented a small fraction of our cohort, and thus we would not expect a substantial variation in the overall conceptual conclusions. This nonetheless is a key design point to integrate into future implementation considerations.

We have added to the discussion section in the paragraph starting with “Positive microbial cultures do not always indicate...” a sentence that notes this limitation.

9. What are the indications for wide spread quinolone use in uncomplicated UTI? This is not something that would normally be considered unless very high rates of ESBL or upper urinary tract infection is suspected / prostate issues? I think that overall, further justification of the agents selected would be beneficial for non-native readers.

Thank you. Despite what one might expect due to criteria stated in the IDSA guidelines, fluoroquinolone use remains high at 40.3% to treat cases of uncomplicated urinary tract infection across the United States¹. Even though guidelines exist that intend to reduce unnecessary use of fluoroquinolones, there is a large amount of variability in the degree to which clinicians adhere to these guidelines potentially due to perceived risk of resistance to narrower-spectrum options like trimethoprim/sulfamethoxazole and variation in risk tolerance amongst practitioners.

We believe this discussion points towards broader-implications of our work, where we have systematically estimated the trade-off in patient coverage when less broad-spectrum antibiotics are used.

We have added text to the first paragraph of the discussion section that calls to the reader's attention high fluoroquinolone use despite IDSA guidelines.

1. S. Kabbani, A. L. Hersh, D. J. Shapiro, K. E. Fleming-Dutra, A. T. Pavia, L. A. Hicks, Opportunities to improve fluoroquinolone prescribing in the United States for adult ambulatory care visits. *Clin. Infect. Dis.* **67**, 134–136 (2018).

From a ML development perspective:

10. In terms of the limitations of predictive models, this study is hampered by similar challenges highlighted in the introduction. Namely the reliance on retrospective data and using culture as a gold standard. The use of ICD-10 codes is not the best source for diagnosis of infection and needs to be acknowledged as a limitation.

We thank the reviewer for bringing up this important discussion point. We agree that the retrospective nature of the study, reliance on microbial cultures for labels and reliance on ICD codes for infection diagnosis in the sensitivity analysis should be acknowledged as limitations. We have subsequently added to our discussion section to make these limitations clear to readers.

Specifically, we discuss the limitation of the retrospective nature of the analysis in the paragraph starting with “We acknowledge limitations associated with the retrospective nature of our analysis...” We discuss the limitation of reliance on ICD codes in the paragraph starting with “Not all microbial cultures return positive...” We acknowledge limitations associated with use of microbial culture results in the paragraph starting with “We acknowledge limitations associated with the use of microbial cultures as a gold standard...”

11. In terms of baseline data how was class imbalance / missingness of data dealt with?

We deliberately allowed real world prevalence (class imbalance) into the model to best preserve calibration in the test set. We did not re-weight or re-sample observations based on class prevalence as this can skew calibration of the predicted probabilities. Missingness was handled implicitly in the featurization scheme — we used a bag of words featurization technique. Each feature was represented by the number of times it occurred in a predefined look back window. If a feature never occurred in that window, the value in the feature matrix was 0. This is particularly helpful in a setting where the fact that data was missing (ex. lack of lab test) can be informative.

We have added a sentence justifying our treatment of class imbalance to the first paragraph in the methods section titled ‘Training and Model Selection Procedure’.

Further we have added a clarifying statement at the end of the first paragraph of the methods section titled “Feature Engineering” that discusses how missingness is handled.

12. It would be interesting to consider what would happen in terms of the predictive value of your algorithms in the face of different prevalence rates of resistance. For example, if the algorithm were deployed in a region with very low / high ESBL or MRSA rates, would this make a large difference? If so would it be a potential consideration for what prescriptions / conditions to target in the future?

We thank the reviewer for this excellent comment. We would expect model calibration to decay to in the presence of data with higher or lower rates of resistance. We would expect measures of discrimination like the area under the receiver operating characteristics curve (AUROC) to remain stable assuming the relationship between features and outcomes does not change. As is commonly the case however, these relationships do change at least moderately from institution to institution, which is why we would expect a model trained on one site to perform more poorly (calibration and discrimination) on data collected from another. Thus it is not our claim that we have trained models that generalize across institutions, but rather that our process of training models and optimizing antibiotic allocations does.

We believe this is an excellent discussion point, and have added commentary in the discussion paragraph starting with “By the nature of differing antibiotic susceptibility patterns...”

REVIEWERS' COMMENTS:

Reviewer #1 (Remarks to the Author):

I want to thank the authors for the well-explained responses to my comments and manuscript revision. There is no further comment.

Reviewer #2 (Remarks to the Author):

Thank you to the authors for comprehensive responses to points made. I am happy with these and this paper well deserves publication.

Reviewer #3 (Remarks to the Author):

The authors have comprehensively addressed the reviewer feedback, clearly and openly acknowledging the limitations of their current work where appropriate.

I note a small typo in table 4 title "taszobactam" that should be corrected during proof-reading.

Otherwise no further feedback.

I wish the authors well and would like to congratulate them on their work. I will follow their progress with interest and hope that they are able to evaluate the tools use in clinical practice in the not too distant future!

Reviewer Comments and Revisions

Referee expertise:

Referee #1: ML, including for antibiotic prescribing

Referee #2: Clinical infectious diseases, ML-guided antibiotic prescribing

Referee #3: Clinical infectious diseases, ML-guided antibiotic prescribing

Legend

Reviewer comments

Author responses

Reviewer #1 (Remarks to the Author):

I want to thank the authors for the well-explained responses to my comments and manuscript revision. There is no further comment.

Reviewer #2 (Remarks to the Author):

Thank you to the authors for comprehensive responses to points made. I am happy with these and this paper well deserves publication.

Reviewer #3 (Remarks to the Author):

The authors have comprehensively addressed the reviewer feedback, clearly and openly acknowledging the limitations of their current work where appropriate.

I note a small typo in table 4 title "taszobactam" that should be corrected during proof-reading.

We thank the reviewer, this typo has been corrected.

Otherwise no further feedback.

I wish the authors well and would like to congratulate them on their work. I will follow their progress with interest and hope that they are able to evaluate the tools use in clinical practice in the not too distant future!

We thank all reviewers for excellent commentary on our manuscript that contributed to a better final product.